# Quantifying alcohol audio-visual content in UK broadcasts of the 2018 Formula 1 Championship: a content analysis and population exposure

Alexander Barker ![ORCID],[1] Magdalena Opazo-Breton,[1] Emily Thomson,[1] John Britton,[2] Bruce Grant-Braham,[3] Rachael L Murray ![ORCID] [1]

¹Division of Epidemiology and Public Health, University of Nottingham, Nottingham, UK
²Division of Epidemiology, University of Nottingham, Nottingham, UK
³Motor Sport Research Group, School of Tourism, Bournemouth University, Bournemouth University, Poole, UK

**Correspondence to**
Dr Alexander Barker;
alexander.barker@nottingham.ac.uk

## ABSTRACT

**Objectives** Exposure to alcohol imagery is associated with subsequent alcohol use among young people, and UK broadcasting regulations protect young people from advertising alcohol content in UK television. However, alcohol promotion during sporting events, a significant potential medium of advertising to children, is exempt. We report an analysis and estimate the UK population exposure to, alcohol content, including branding, in UK broadcasts of the 2018 Formula 1 (F1) Championship.

**Setting** UK.

**Participants** None. Content analysis of broadcast footage of 21 2018 F1 Championship races on Channel 4, using 1-minute interval coding of any alcohol content, actual or implied use, other related content or branding. Census data and viewing figures were used to estimate gross and per capita alcohol impressions.

**Results** Alcohol content occurred in all races, in 1613 (56%) 1-minute intervals of race footage and 44 (9%) of intervals across 28% of advertisement breaks. The most prominent content was branding, occurring in 51% of race intervals and 7% of advertisement break intervals, appearing predominantly on billboard advertisements around the track, with the Heineken and Johnnie Walker brands being particularly prominent. The 21 races delivered an estimated 3.9 billion alcohol gross impressions (95% CI 3.6 to 4.3) to the UK population, including 154 million (95% CI 124 to 184) to children, and 3.6 billion alcohol gross impressions of alcohol branding, including 141 million impressions to children. Branding was also shown in race footage from countries where alcohol promotion is prohibited.

**Conclusions** Alcohol content was highly prevalent in the 2018 F1 Championship broadcasts, delivering millions of alcohol impressions to young viewers. This exposure is likely to represent a significant driver of alcohol consumption among young people.

## INTRODUCTION

Alcohol consumption in the UK is the eighth highest in Europe[1] and currently causes at least 7000 UK deaths[2] and an estimated economic burden of between £21 and £52 billion each year.[3] It is estimated that 1.6 million people in the UK have some

### Strengths and limitations of this study

► This study builds on previous studies by estimating population exposure to alcohol content shown in Formula1 (F1) races.
► Established methods were used to explore alcohol content in the 2018 F1 Championship races.
► This study only explored broadcasts where races occurred and not supplementary programmes.
► Future studies should explore sales data to examine whether this exposure led to an increase in sales.

form of alcohol dependence;[4] alcohol misuse increases the risk of serious health conditions such as stroke, heart disease and cancer.[5] Preventing alcohol morbidity and mortality is, therefore, a clear public health priority.

There is strong evidence that exposure to advertising or other alcohol audio-visual content (AVC) in the media increases subsequent alcohol use by adolescents,[6–9] and both commercial advertising of alcohol and alcohol content in broadcast programmes are regulated in the UK to protect adolescents from exposure. However, Advertising Standards Authority commercial advertising regulatory codes do not cover broadcast footage of imagery arising from sponsorship of sporting events;[10 11] since this is programme content, it is regulated instead by the Office of Communications (Ofcom). The Ofcom Broadcasting Code[12] restricts depictions of alcohol misuse in programmes made for children, and the glamorisation of alcohol use in programmes broadcast before the 9 pm watershed[13] otherwise likely to be widely seen, heard or accessed by children without editorial justification (Section 1.10). The Code also restricts advertising within programmes, stating that 'no undue prominence may be given in programming to a product, service or trademark' through the

presence of reference to a product where there is no editorial justification or the manner in which a product, service or trademark appears in programming (Section 9.5). However, Ofcom has no remit over sports sponsorship deals, such as a company sponsoring a stadium, a team or an individual sportsperson, and Ofcom guidance states that the context of advertising is taken into account with a greater number of in situ advertising expected at sporting venues.[14 15] Alcohol advertising during televised sporting events is thus a potentially unregulated source of exposure to alcohol content for young people.

While our previous research has shown that alcohol content and branding is highly prevalent in a sample of Formula 1 (F1) Championship races,[16] the total amount of content and branding shown through an entire annual championship, and the population exposure (particularly among young people) to this content remains unknown. In 2018, Heineken continued to sponsor the F1 Championship races[17] which were viewed by 45.5 million viewers in the UK,[18] an estimated 14% of which were aged under 25 years.[19] We have therefore quantified alcohol AVC in all 21 races from the 2018 F1 Championship and estimated UK population exposure, including child exposure, to this content. We have also compared the amount of alcohol imagery in the 2018 race series with our earlier analysis of footage from the 2017 competition.

## METHODS

The 2018 F1 Championship took place between 15 March 2018 and 1 December 2018 and featured races in 21 countries. Ten of these races were broadcast in the UK live and in full by Channel 4, and 11 in selected highlight form. These broadcasts were recorded in their entirety (start of the programme until the end of the credits), viewed and coded using the 1-minute interval period method previously described,[16 20–23] coding the presence of audio-visual alcohol content every 1-minute interval in the following categories:

*Actual use*: Use of alcohol onscreen by any person (eg, seeing a person actually consume alcohol on screen).

*Implied use*: Any implied alcohol use without any actual use on screen (eg, seeing a person holding a drink/bottle of alcohol, but not actually consuming alcohol).

*Other alcohol reference*: The presence onscreen of alcohol or other related materials (eg, bottles or beer pumps not currently in use, or advertising materials).

*Brand appearance*: The presence of clear and unambiguous branding (eg, when a brand is identified on screen). Brand appearances were divided into the following categories; alcoholic and 0% or low alcohol products, products which, while not an alcohol product, share branding with alcoholic products.

*Any alcohol content*: Any of the above.

One-minute intervals were used rather than shorter intervals used in other studies,[24] due to the length of the programme being coded and for practical issues around the amount of time required to code in 10-second intervals.

For coding purposes, multiple instances of the same category in the same 1-minute period were considered to be a single event, while instances that ran into consecutive 1-minute periods were coded as separate events. Instances in different categories in the same interval were recorded as different events. The alcohol content was recorded in two categories in the same interval if the content met the criteria for two categories, for example, if a driver drank from a branded bottle, this would be recorded as actual use and a brand appearance. Ten per cent of the coding was checked by a second author (ET) to ensure accuracy and reliability in the coding method.

We estimated UK audience exposure using viewing data from Digital.I[25] and UK mid-year population estimates for 2017[26] combined with numbers of alcohol appearances (every minute containing alcohol use, implied use, other references or branding) to estimate gross (the total number of impressions delivered to the UK population) and per capita (the number of impressions delivered to each person) 1-minute impressions by age group, using previously reported methods.[27–29] The method involves combining viewership (obtained from viewing figures) with the number of intervals containing alcohol appearances per episode to calculate gross impressions as the estimated number of exposures delivered. Dividing gross impressions by population mid-year estimates provided per capita impressions, the estimated number of alcohol impressions delivered to each person. Both gross and per capita impressions were computed by age group. Analyses were conducted in IBM SPSS Statistics (V.24) and Microsoft Excel (2013). The confidence level was set to 95%. We compared content with findings of our earlier studies alcohol content in a sample of F1 races from 2017[16] using $\chi^2$ analysis.

### Patient and public involvement

No patients were involved in this study.

## RESULTS

We coded a total of 3396 1-minute intervals, of which 2899 were from race footage and 497 from advertisement breaks, all of which were broadcast by Channel 4 on a Sunday, with start times ranging from 06:00 to 23:00 and a mean broadcast duration (including advertisement breaks) of 2 hours 21 min (range=1 hour 34 min–3 hours 7 min). The inter-rater coding agreement was 98.98% of 1-minute intervals in the sample double coded, with differences, resolved on discussion and multiple viewings.

### Any alcohol content

In total, 1613 (56%) 1-minute intervals of racing and 44 (9%) 1-minute advertisement intervals contained any alcohol content, with occurrences in all 21 races and 28% of advertisement breaks (table 1). The majority of alcohol content occurred during race footage.

### Actual use

Alcohol consumption during race footage was limited to consumption of a sparkling drink on the winner's

| Table 1 Alcohol AVC content by type and location | | |
|---|---|---|
| Type of content | During race broadcast N (%) | During advertisements N (%) |
| Any alcohol content | 1613 (56) | 44 (9) |
| Actual use | 27 (1) | 2 (2) |
| Implied use | 92 (3) | 34 (7) |
| Other alcohol references | 1571 (54) | 38 (8) |
| Brand appearance | 1490 (51) | 37 (7) |
| Heineken branding | 1029 (35) | 2 (2) |

AVC, audio-visual content.

podium, identified from branding on the bottle or podium as Carbon Champagne, which occurred at the end of 18 races (86%) in a total of 25 (1%) 1-minute intervals. There were 10 advertisement breaks depicting alcohol consumption, across 11 1-minute intervals (<1%).

## Implied use

Implied alcohol use was observed in 92 (3%) race intervals, predominantly the sparkling drink being sprayed from the winner's podium. Implied use was seen in 34 1-minute intervals during advertisement breaks, and predominantly comprised imagery of people holding drinks during advertisements for alcohol brands.

## Other alcohol references

Alcohol references, predominantly involving billboard and car brand advertising, occurred in 1571 race intervals (46% of all race intervals), including 'When you drink, never drive' billboards, which appeared in 241 (7%) of race intervals. Alcohol references also occurred in 38 (8%) 1-minute intervals from advertisement breaks, mostly consisting of bottles or branded items (such as beer pumps) (table 1).

## Brand appearances

Alcohol product branding in billboard advertisements or on the side of cars or racing suits occurred in 1490 (51%) race intervals across all 21 races. The most common brands observed were Heineken, which occurred in 1031 intervals and appeared 8795 times; and Johnnie Walker, which appeared in 608 intervals and appeared on screen 4364 times. Heineken 0.0, a non-alcoholic lager which shared the Heineken brand name and imagery, occurred in 120 1-minute intervals (8% of the 1490 intervals containing branding) and appeared 548 times. Alcohol branding occurred in 37 (7%) advertisement break intervals, with the most common brand being Heineken (table 1). A list of all the brands observed is presented in figure 1.

In addition to billboard advertising, computer-generated advertising added to highlights programmes were seen in 17 intervals during two races (Canada and Austria). These intervals featured a red star, reminiscent of the Heineken logo, superimposed over the track.

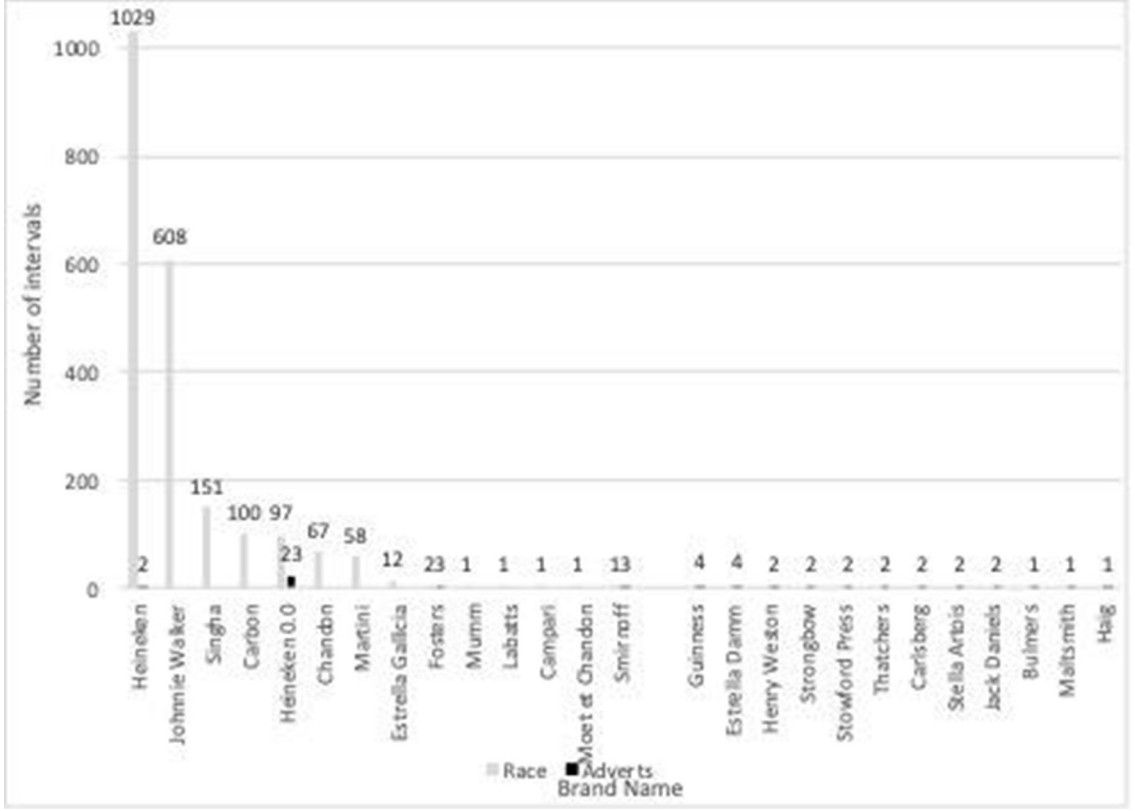

**Figure 1** Brands identified in race footage and advertisements.

## Differences between race venues

The amount of alcohol content and alcohol branding was highest in Monaco GP and lowest in the Bahrain GP (table 2).

## UK population exposure

We estimate that the 21 race broadcasts delivered 3.9 billion alcohol gross impressions (95% CI 3.6 to 4.3) to the UK population, including 154 million (95% CI 124 to 184) to children aged under 16 years. Alcohol impressions per capita were highest (average 513.11 (95% CI 499.96 to 526.25)) in the 55–64 year age group and were higher in men than women (513.11 (95% CI 499.96 to 526.25) and (433.50 (95% CI 420.98 to 446.02), respectively). Children received an average 12.25 (95% CI 9.84 to 14.66) per capita impressions. Of these impressions, 3.6 billion (95% CI 3.3 to 4.0) were of alcohol branding and including 141 million (95% CI 113 to 169) to children. (see online supplementary table 1 for a breakdown of gross and per capita impressions per race). We estimate that the computer-generated branding features in 2 races delivered 269.34 million gross impressions to the UK population, including 1.69 million to children.

## Comparison to 2017 Championship

Comparison with our earlier analysis of alcohol content in the 2017 F1 Championship[16] demonstrated significantly greater alcohol content in the 2018 season, particularly in relation to the proportion of intervals including alcohol branding, and especially Heineken advertising (table 3). Heineken 0.0 advertising was significantly less prevalent in the 2018 footage (table 3). In our previous content analysis,[16] Heineken 0.0 branding was seen at only two events, the Spanish Grand Prix and the Monaco Grand Prix. In the 2018 Monaco Grand Prix, billboard advertising for Heineken 0.0 had been replaced with conventional Heineken imagery.

## DISCUSSION

This study demonstrates that audio-visual alcohol content, including branding and particularly Heineken and Johnnie Walker branding, was highly prevalent in broadcast footage of the 2018 F1 Championship races, and that the overall amount of alcohol content in 2018 was greater than in 2017,[16] largely because of substitution of zero-alcohol with alcoholic Heineken branding. The alcohol content in the broadcast race footage generates substantial population exposure, delivering billions of viewer impressions. These included more than 100 million gross branded alcohol impressions to children aged under 16 years.

Our study thus provides evidence that UK broadcast footage of the F1 Championship is a significant source of exposure to alcohol content and advertising for children. The alcohol content was prevalent in all races, even races which took place in countries with alcohol advertising restrictions. Alcohol advertising, through sponsorship deals, during these events appears to be changing with the inclusion of computer-generated alcohol advertising, which was introduced in the 2018 Championship. Since the F1 Championships are international events with global audiences, the UK population exposure figures likely represent a very small proportion of the true total global exposure.

Some alcohol producers associate their advertising with responsible drinking messages. Heineken's 5-year F1 sponsorship has been heavily criticised for 'linking a popular motorsport to a significant cause of avoidable physical, mental and social harm and more specifically one of the major killers on our roads, drink driving'.[30] This global campaign incorporates Heineken's characteristic red star and green branding on billboards which also have a prominent 'When You Drive, Never Drink' message.[31] Indeed Heineken has said that it will use F1 to promote this campaign[32] and the Heineken 0.0 brand.[33] We have previously argued that Heineken 0.0 advertising may be acting as an alibi for alcohol products,[16] not least because they share the same brand imagery (a star in the case of Heineken) on both alcohol and low alcohol products, potential consumers, and particularly young people, may well not distinguish advertising for alcoholic and non-alcoholic Heineken products. This industry tactic could potentially be serving a subtle public relations function, engendering goodwill with potential consumers of the brand while increasing the promotion of their alcohol products, promoting brand preference and product consumption.[34 35] The current study has identified that the proportion of Heineken 0.0 advertising has decreased significantly since our 2017, with billboards for Heineken 0.0 from 2017 being replaced with regular Heineken billboards.

The majority of the F1 Championship races (18 of 21) were broadcast on a Sunday before the 9 pm watershed, a time when children are likely to be watching what their parents are watching.[36] Channel 4 state that they 'abide' by the Ofcom Broadcasting Code,[12] which considers factors that determine whether a programme should be shown including 'the likely number and age range of children watching, taking into account school time, weekends and holidays'.[37] The current study has demonstrated that the broadcast footage of the F1 Championship delivers millions of branded alcohol content impressions to children. Furthermore, the Ofcom code also states that 'Before the watershed, …the misuse of alcohol must never be condoned, encouraged or glamorised and scenes showing such material should generally be avoided unless there is editorial justification'.[12] All of the F1 races featured scenes of drivers celebrating their win by drinking from and spraying bottles of champagne, a traditional celebration in F1,[38] which could potentially be associating driving with alcohol use[39] and glamorising alcohol use by showing drivers consuming alcohol in what F1's Chief Executive Officer considers to be a glamourous sport.[40]

Rule 9.5 of the Ofcom Broadcasting Code[12] states that 'no undue prominence may be given in programming to

**Table 2** Comparison of alcohol content between race venues (in order of broadcast)

Number of 1-minute intervals (% proportion of race intervals)

| | Australia | Bahrain | China | Azerbaijan | Spain | Monaco | Canada | France | Austria | Britain | Germany | Hungary | Belgium | Italy | Singapore | Russia | Japan | USA | Mexico | Brazil | Abu Dhabi |
|---|---|---|---|---|---|---|---|---|---|---|---|---|---|---|---|---|---|---|---|---|---|
| Any alcohol content | 79 (55) | 18 (12) | 92 (73) | 81 (43) | 48 (45) | 129 (80) | 90 (96) | 15 (15) | 82 (49) | 110 (66) | 65 (55) | 55 (48) | 117 (74) | 88 (76) | 122 (71%) | 40 (34%) | 75 (50%) | 84 (55%) | 97 (83%) | 102 (87%) | 24 (15%) |
| Alcohol use | 1 (1) | 0 (0) | 1 (1) | 1 (1) | 2 (2) | 3 (2) | 1 (1) | 2 (2) | 2 (1) | 1 (1) | 1 (1) | 1 (1) | 2 (1) | 1 (1) | 1 (1) | 2 (2) | 1 (1) | 1 (1) | 1 (1) | 1 (1) | 1 (1) |
| Implied use | 5 (1) | 1 (1) | 2 (2) | 4 (2) | 7 (7) | 5 (3) | 5 (5) | 4 (4) | 2 (1) | 2 (1) | 4 (3) | 5 (4) | 2 (1) | 6 (5) | 0 (0) | 5 (4) | 1 (1) | 5 (3) | 13 (11) | 10 (9) | 4 (2) |
| Other alcohol references | 76 (53) | 13 (9) | 92 (73) | 77 (41) | 43 (41) | 126 (78) | 90 (96) | 15 (15) | 80 (48) | 109 (66) | 63 (53) | 53 (46) | 117 (74) | 87 (75) | 121 (70) | 38 (33) | 75 (50) | 82 (54) | 91 (78) | 102 (87) | 21 (13) |
| Brand appearance | 59 (41) | 10 (7) | 92 (73) | 68 (36) | 39 (37) | 121 (75) | 84 (89) | 11 (11) | 79 (47) | 108 (66) | 62 (53) | 48 (42) | 114 (72) | 87 (75) | 121 (70) | 36 (31) | 63 (42) | 82 (54) | 89 (76) | 101 (86) | 16 (10) |
| Any alcohol content | 79 (55) | 18 (12) | 92 (73) | 81 (43) | 48 (45) | 129 (80) | 90 (96) | 15 (15) | 82 (49) | 110 (66) | 65 (55) | 55 (48) | 117 (74) | 88 (76) | 122 (71) | 40 (34) | 75 (50) | 84 (55) | 97 (83) | 102 (87) | 24 (15) |
| Alcohol use | 1 (1) | 0 (0) | 1 (1) | 1 (1) | 2 (2) | 3 (2) | 1 (1) | 2 (2) | 2 (1) | 1 (1) | 1 (1) | 1 (1) | 2 (1) | 1 (1) | 1 (1) | 2 (2) | 1 (1) | 1 (1) | 1 (1) | 1 (1) | 1 (1) |
| Implied use | 5 (1) | 1 (1) | 2 (2) | 4 (2) | 7 (7) | 5 (3) | 5 (5) | 4 (4) | 2 (1) | 2 (1) | 4 (3) | 5 (4) | 2 (1) | 6 (5) | 0 (0) | 5 (4) | 1 (1) | 5 (3) | 13 (11) | 10 (9) | 4 (2) |
| Other alcohol references | 76 (53) | 13 (9) | 92 (73) | 77 (41) | 43 (41) | 126 (78) | 90 (96) | 15 (15) | 80 (48) | 109 (66) | 63 (53) | 53 (46) | 117 (74) | 87 (75) | 121 (70) | 38 (33) | 75 (50) | 82 (54) | 91 (78) | 102 (87) | 21 (13) |
| Brand appearance | 59 (41) | 10 (7) | 92 (73) | 68 (36) | 39 (37) | 121 (75) | 84 (89) | 11 (11) | 79 (47) | 108 (66) | 62 (53) | 48 (42) | 114 (72) | 87 (75) | 121 (70) | 36 (31) | 63 (42) | 82 (54) | 89 (76) | 101 (86) | 16 (10) |
| Heineken | 33 (23) | 0 (0) | 85 (67) | 59 (32) | 0 (0) | 114 (70) | 82 (87) | 1 (1) | 69 (41) | 82 (49) | 14 (12) | 40 (35) | 44 (28) | 74 (64) | 86 (50) | 1 (1) | 65 (43) | 12 (8) | 71 (60) | 96 (82) | 1 (1) |
| Heineken 0.0 | 0 (0) | 0 (0) | 0 (0) | 0 (0) | 30 (28) | 0 (0) | 0 (0) | 0 (0) | 0 (0) | 0 (0) | 0 (0) | 0 (0) | 0 (0) | 36 (31) | 0 (0) | 31 (26) | 0 (0) | 0 (0) | 0 (0) | 0 (0) | 0 (0) |

**Table 3** Proportion (%) of race footage containing alcohol content in the 2018 and 2017 Championships

| | 2018 Championship (%) | 2017 Championship (%) | P value |
|---|---|---|---|
| **Alcohol** | | | |
| Any alcohol content | 56 | 46 | <0.01 |
| Actual alcohol use | 1 | 1 | 1 |
| Implied alcohol use | 3 | 4 | 0.1498 |
| Other alcohol reference | 54 | 44 | <0.01 |
| Alcohol branding | 51 | 35 | <0.01 |
| Heineken | 35 | 13 | <0.01 |
| Heineken 0.0 | 3 | 14 | <0.01 |

a product, service or trademark' through the presence of or reference to a product where there is no editorial justification or the manner in which a product, service or trademark appears in programming. While this rule takes into account the context and in situ advertising at venues is editorially justified, the current study found that computer-generated Heineken branding was added to F1 race highlights programmes, through the inclusion of a red star superimposed over the track similar to Heineken's red star logo representing an apparent clear breach of the Ofcom Broadcasting Code. Channel 4 was recently found to be in breach of the code for their F1. We reported this to Ofcom[15] which noted that the two Grands Prix containing computer-generated alcohol advertising were broadcast before an Ofcom ruling which found Channel 4 to be in breach of the code for their F1 coverage featuring computer-generated 'Rolex' branding (a Rolex clock face superimposed over footage of The Singapore Flyer, a large Ferris wheel),[41] and expect broadcasters to take the precedent of the Rolex ruling into account for subsequent broadcasts.

While there is scope under Ofcom's powers to regulate content in programmes broadcast in the UK, Ofcom has no remit over sports sponsorship deals, and these are therefore a potentially unregulated source of exposure to alcohol content for young people. Furthermore, the F1 Championship races are broadcast internationally, which complicates regulating this content. The Framework Convention for Tobacco Control[42] is a treaty negotiated under the auspices of the WHO which provides a regulatory strategy to address addictive substances, has placed an international, comprehensive ban on the advertising, promotion and sponsorship of tobacco products.[43] A similar comprehensive ban on the advertising, promotion and sponsorship of alcohol products would prevent young people from being exposed to this currently unregulated alcohol promotion.

Alcohol content and brand prevalence differed between race venues, possibly due to differences in national restrictions on alcohol advertising.[44] The races with the least alcohol content were Bahrain, Abu Dhabi and France; Bahrain and Abu Dhabi are events in Islamic countries where alcohol advertising is prohibited, while France has statutory legislation, 'the Loi Evin',[45] prohibiting alcohol marketing and advertising by placing a total ban on the direct or indirect advertising of all alcoholic beverages over 1.2% ABV (Alcohol by Volume) on television and prohibiting sponsorship of sports events by alcohol companies.[46] Alcohol advertising did occur in UK broadcast footage of races in countries where alcohol advertising is prohibited, such as Russia, France and Bahrain, raising concerns that alcohol brands are bypassing country-specific regulations in the F1 races. Race footage featuring billboard advertisements are also broadcast worldwide, which could circumvent alcohol advertising regulations in a similar way to the UK's lack of regulation covering advertisements at sporting venues.[10 11 14 15]

Our study is limited to UK broadcasts shown on Channel 4, and it is possible that other broadcasts, such as those shown on Sky one, showed different footage. The current quantification and population exposure only explored UK footage, and therefore is unable to estimate the amount or exposure to content broadcast in other countries; however, it is likely that our findings apply much more widely than in the UK. Channel 4 also broadcast qualifying race and pre-event programmes and postevent programmes, in this content analysis, we focused on the broadcast containing the actual race as specified by Channel 4, because of this it is likely that this content analysis underestimates the amount of alcohol AVC shown in the F1 Championship.

While the current study found that alcohol branding was highly prevalent in broadcast footage of the 2018 F1 Championship races, we currently do not know whether this led to increased alcohol sales. Future studies should explore sales data for prevalent brands to identify if the presence of branding in such events led to an increase in sales of branded goods featured in the championship races. Channel 4 also broadcast other F1 programmes, including qualifying race and pre-event programmes and postevent programmes, in this content analysis, we focused on the broadcast containing the actual race as specified by Channel 4, rather than qualifying or postevent programmes. It is therefore likely that this content analysis underestimates the amount of alcohol AVC shown in the F1 Championship as a whole, and the population exposure to this content.

## CONCLUSION

Alcohol content shown on TV, including alcohol advertisements and branding, is known to have an effect on the uptake of alcohol use in young people.[47] Alcohol content broadcast during the F1 races resulted in millions of gross alcohol content impressions being delivered to children,

and this exposure is likely to lead to subsequent alcohol use. We also found computer-generated branding in the F1 races broadcast in the UK, contravening an earlier Ofcom ruling, and alcohol advertising in countries where this is prohibited. Restrictions on and enforcement of alcohol AVC during sporting events are needed to protect children and adolescents from this avenue of alcohol advertising.

**Acknowledgements** JB and RLM are members of SPECTRUM, a UK Prevention Research Partnership Consortium. UKPRP is an initiative funded by the UK Research and Innovation Councils, the Department of Health and Social Care (England) and the UK devolved administrations, and leading health research charities.

**Contributors** AB led the data coding and data analysis and contributed to the manuscript. ET double coded the data to ensure reliability. MO-B supported data analysis. BG-B contributed with interpreting the findings of the study. JB and RLM contributed to drafting the manuscript. All authors read and approved the final manuscript.

**Funding** This work was supported by the Medical Research Council (grant number MR/K023195/1) and the UK Centre for Tobacco and Alcohol Studies, with core funding from the British Heart Foundation, Cancer Research UK, Economic and Social Research Council and the Department of Health under the auspices of the UK Clinical Research Collaboration. The funders had no role in the study design, data collection and analysis, decision to publish or preparation of the manuscript.

**Competing interests** None declared.

**Patient consent for publication** Not required.

**Provenance and peer review** Not commissioned; externally peer reviewed.

**Data availability statement** Data are available upon reasonable request. Availability of data and materials: Data and materials are available from the lead author upon reasonable request.

**ORCID iDs**
Alexander Barker http://orcid.org/0000-0003-4568-5114
Rachael L Murray http://orcid.org/0000-0001-5477-2557

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
