## [Reviewer comments · BMJ Open]

ARTICLE DETAILS

TITLE (PROVISIONAL)	Quantifying alcohol audio-visual content in UK broadcasts of the 2018 Formula 1 Championship: a content analysis and population exposure.
AUTHORS	Barker, Alexander; Opazo-Breton, Magdalena; Thomson, Emily; Britton, John; Grant-Braham, Bruce; Murray, Rachael

VERSION 1 – REVIEW

REVIEWER	Pamela Trangenstein University of North Carolina at Chapel Hill
REVIEW RETURNED	17-Feb-2020

GENERAL COMMENTS	This study examined the gross and per capita impressions of alcohol advertising during the 2018 Formula 1 Championship (comprising 21 races) broadcast in the UK in 2018. It is an update of a previous study that examined the first six races of the F1 Championship in 2017. The biggest changes from 2017 to 2018 appear to be the addition of computer-generated advertisements at two stadiums and replacing billboards advertising non-alcoholic Heineken products with alcoholic products at up to two stadiums. Overall, this manuscript helps identify ways that alcohol advertising may reach young viewers in the UK despite the current restrictions. Major concerns 1. In its current form, the significance of this manuscript over the previous analysis is not clear. It appears that the authors used the previous manuscript as a template and updated some words/sections. The core arguments appear largely the same in the two manuscripts; please clarify the unique contribution of this analysis over and above the previous manuscript.2. The manuscript is missing a limitations section. Minor concerns Introduction 3. It would be helpful to provide statistics on the demographics of the viewers of F1 Championship races in the UK, particularly with regard to age.4. If Heineken continued to sponsor the F1 Championship races, this is important to note in the introduction or methods section to help readers interpret the findings about branding. Methods 5. Why was the 2017 population midpoint used to calculate per capita impressions when the races were broadcast between March and December 2018?6. Please add a justification for why the one-minute interval coding was preferable to shorter intervals (e.g., 10 seconds) in this study.
---

	7. The codes used in this study appear to change the alcohol paraphernalia category to "other alcohol content" to include billboards and advertisements on cars from the cited references (although it was used in the 2017 analysis). What was the justification for coding these background advertisements combined with paraphernalia instead of coded as a new category? It is also not clear that spraying alcohol on crowds fits the provided definition of "implied use." Please clarify this fit. 8. The definitions of key terms (e.g., impressions, gross impressions, per capita impressions) are all present but could be hard to understand for audiences who are unfamiliar with them. It may help to define the terms before describing how they were calculated. Results 9. Please report the results of the inter-rater coding agreement for the subset of intervals that were coded by two coders. Discussion 10. The term "alcohol advertising" is too broad to permit a focused discussion and clear set of recommendations. Please consider clarifying the medium (e.g., television, billboards), type of advertising (e.g., promotions, sponsorship), and/or type of regulation (e.g., volume, placement, content) throughout the discussion. 11. Can the authors go further in their recommendations? For example, does the international context of the races present unique challenges in regulating alcohol content exposure? Who is responsible for enforcing the OfCom 'Rolex' ruling, and what could be done to ensure it is followed? 12. The statement in lines 12-13 on page 11 "the current study demonstrated that the broadcast footage of the F1 Championship is widely seen by children..." is not a product of this study; it was demonstrated by the Digital.I data. Perhaps the authors meant that the study estimated the volume of youth alcohol advertising impressions during this footage?
--	---

REVIEWER	Frank Houghton Limerick Institute of Technology, Limerick, Ireland
REVIEW RETURNED	24-Feb-2020

GENERAL COMMENTS	A useful study. Just a few minor points: I presume in the abstract it should include the word races after '21 2018 F1 Championship' Perhaps after mentioning the number of deaths and economic cost of alcohol a sentence on morbidity associated with it might be of use. I am a little unclear what a gross alcohol impression exactly means here. I know you reference the methodology elsewhere, but a little more clarity here would help. Does it mean alcohol use/ implied use/ advertised etc once in a minute? What exact time frame is under analysis? From the start flag to the final flag? Or 10 mins either side? Or the whole program? Or the start flag to the award of the trophy? Might it be worth suggesting an alcohol equivalent of the FCTC?
---

	Good Work- To be Commended
--	----------------------------

VERSION 1 – AUTHOR RESPONSE

Reviewer: 1

Reviewer Name: Pamela Trangenstein

Institution and Country: University of North Carolina at Chapel Hill

Please state any competing interests or state 'None declared': None declared

Please leave your comments for the authors below

This study examined the gross and per capita impressions of alcohol advertising during the 2018 Formula 1 Championship (comprising 21 races) broadcast in the UK in 2018. It is an update of a previous study that examined the first six races of the F1 Championship in 2017. The biggest changes from 2017 to 2018 appear to be the addition of computer-generated advertisements at two stadiums and replacing billboards advertising non-alcoholic Heineken products with alcoholic products at up to two stadiums. Overall, this manuscript helps identify ways that alcohol advertising may reach young viewers in the UK despite the current restrictions.

Major concerns

1. In its current form, the significance of this manuscript over the previous analysis is not clear. It appears that the authors used the previous manuscript as a template and updated some words/sections. The core arguments appear largely the same in the two manuscripts; please clarify the unique contribution of this analysis over and above the previous manuscript.

We have added the following sentence to explain how this paper builds on the previous paper.

'Whilst our previous research has shown that alcohol content and branding is highly prevalent in a sample of F1 Championship races (14), the total amount of content and branding shown through an annual championship, and the population exposure (particularly among young people) to this content remains unknown.' – Page 4, line 19-22

2. The manuscript is missing a limitations section.

A limitations section has been added.

'Our study is limited to UK broadcasts shown on Channel 4, it is possible that other broadcasts, such as those shown on Sky one, showed different footage. The current quantification and population exposure only explored UK footage, and therefore is unable to estimate the amount or exposure to content broadcast in other countries, however, it is likely that our findings apply much more widely than in the UK. Channel 4 also broadcast qualifying race and pre and post-event programmes, in this content analysis we focused on the broadcast containing the actual race as specified by Channel 4, because of this it is likely that this content analysis underestimates the amount of alcohol AVC shown in the F1 Championship.' – Page 16, line 20-Page 17, line 5.

Minor concerns

Introduction

3. It would be helpful to provide statistics on the demographics of the viewers of F1 Championship races in the UK, particularly with regard to age.

We have added the following sentence to the introduction.

'In 2018, Heineken continued to sponsor the F1 Championship races (17) which were viewed by 45.5 million viewers in the UK (18), an estimated 14% of which were aged under 25 (19).' – Page 4, line 22 – Page 5, line 1.

4. If Heineken continued to sponsor the F1 Championship races, this is important to note in the introduction or methods section to help readers interpret the findings about branding.

We have now clarified that Heineken continued to sponsor the F1 Championship races in 2018

'In 2018, Heineken continued to sponsor the F1 Championship races (17) which were viewed by 45.5 million viewers in the UK (18), an estimated 14% of which were aged under 25 (19).' – Page 4, line 22 – Page 5, line 1.

Methods

5. Why was the 2017 population midpoint used to calculate per capita impressions when the races were broadcast between March and December 2018?

These were the latest available figures when this analysis was conducted. These have now been updated to the 2018 mid-point statistics and all exposure figures have been updated.

6. Please add a justification for why the one-minute interval coding was preferable to shorter intervals (e.g., 10 seconds) in this study.

'One-minute intervals were used rather than shorter intervals used in other studies (24), due to the length of the programme being coded and for practical issues around the amount of time required to code in 10 second intervals.' – Page 6, line 4-6

7. The codes used in this study appear to change the alcohol paraphernalia category to "other alcohol content" to include billboards and advertisements on cars from the cited references (although it was used in the 2017 analysis). What was the justification for coding these background advertisements combined with paraphernalia instead of coded as a new category? It is also not clear that spraying alcohol on crowds fits the provided definition of "implied use." Please clarify this fit.

We used the same coding categories for this study as we did our previous study, advertising billboards and brand names on cards were coded as 'Other alcohol reference' in line with our previous F1 study and other studies looking at alcohol and tobacco content in the media.

We have defined implied alcohol use as 'Any implied alcohol use without any actual use on screen (for example, seeing a person holding a drink/bottle of alcohol, but not actually consuming alcohol)', in line with our previous work. The drivers on the podium were mostly seen spraying each other before consuming the alcohol. As the driver had opened the bottle before spraying it in other drivers faces, usually before drinking from it themselves, we decided to include the presence of an open spraying bottle as implied use

8. The definitions of key terms (e.g., impressions, gross impressions, per capita impressions) are all present but could be hard to understand for audiences who are unfamiliar with them. It may help to define the terms before describing how they were calculated.

We have now clarified these terms in the text

'We estimated UK audience exposure using viewing data from Digital.I (19) and UK mid-year population estimates for 2017 (20) combined with numbers of alcohol appearances (every minute containing alcohol use, implied use, other references or branding) to estimate gross (the total number of impressions delivered to the UK population) and per capita (the number of impressions delivered to each person) one-minute impressions by age group, using previously reported methods (21-23).' – Page 6, Line 14-19

Results

9. Please report the results of the inter-rater coding agreement for the subset of intervals that were coded by two coders.

This has now been added to the results.

'Inter-rater coding agreement was 98.98% of 1-minute intervals in the sample double coded, with differences resolved upon discussion and multiple viewings.' – Page 7, line 14-15.

Discussion

10. The term "alcohol advertising" is too broad to permit a focused discussion and clear set of recommendations. Please consider clarifying the medium (e.g., television, billboards), type of advertising (e.g., promotions, sponsorship), and/or type of regulation (e.g., volume, placement, content) throughout the discussion.

We have now clarified that we are referring to sports sponsorship deals when we mention 'alcohol advertising' in the discussion.

11. Can the authors go further in their recommendations? For example, does the international context of the races present unique challenges in regulating alcohol content exposure? Who is responsible for enforcing the OfCom 'Rolex' ruling, and what could be done to ensure it is followed?

The following has now been added to the discussion section where we argue that international regulation is needed.

'Whilst there is scope under Ofcom's powers to regulate content in programmes broadcast in the UK, Ofcom has no remit over sports sponsorship deals, and these are therefore a potentially unregulated source of exposure to alcohol content for young people. Furthermore, the F1 Championship races are broadcast internationally, which complicates regulating this content. The Framework Convention for Tobacco Control (42) is a treaty negotiated under the auspices of the World Health Organisation which provides a regulatory strategy to address addictive substances, has placed an international, comprehensive ban on the advertising, promotion and sponsorship of tobacco products (43). A similar comprehensive ban on the advertising, promotion and sponsorship of alcohol products would prevent young people being exposed to this currently unregulated alcohol promotion.' – Page 15, line 20 – Page 16, line 6.

12. The statement in lines 12-13 on page 11 "the current study demonstrated that the broadcast footage of the F1 Championship is widely seen by children..." is not a product of this study; it was demonstrated by the Digital.I data. Perhaps the authors meant that the study estimated the volume of youth alcohol advertising impressions during this footage?

That is correct. The sentence has now been changed from 'The current study has demonstrated that the broadcast footage of the F1 Championship is widely seen by children and delivers millions of branded alcohol content impressions to children' to 'The current study has demonstrated that the broadcast footage of the F1 Championship delivers millions of branded alcohol content impressions to

children'. – Page 14, line 18-20

Reviewer: 2

Reviewer Name: Frank Houghton

Institution and Country: Limerick Institute of Technology,
Limerick,
Ireland

Please state any competing interests or state 'None declared': None declared

Please leave your comments for the authors below

A useful study. Just a few minor points:

I presume in the abstract it should include the word races after '21 2018 F1 Championship'

Yes, this was a mistake and has now been added to the abstract

Perhaps after mentioning the number of deaths and economic cost of alcohol a sentence on morbidity associated with it might be of use.

The following sentence has now been added.

'It is estimated that 1.6 million people in the UK have some form of alcohol dependence (4), alcohol misuse increases the risk of serious health conditions such as stroke, heart disease and cancer (5)' – Page 3, line 20-22.

I am a little unclear what a gross alcohol impression exactly means here. I know you reference the methodology elsewhere, but a little more clarity here would help. Does it mean alcohol use/ implied use/ advertised etc once in a minute?

We have added in details to our methods section to clarify our method of calculating impressions and what we mean by impressions.

“We estimated UK audience exposure using viewing data from Digital.I (19) and UK mid-year population estimates for 2017 (20) combined with numbers of alcohol appearances (every minute containing alcohol use, implied use, other references or branding) to estimate gross (the total number of impressions delivered to the UK population) and per capita (the number of impressions delivered to each person) one-minute impressions by age group, using previously reported methods (21-23).’ – Page 6, Line 14-19

What exact time frame is under analysis? From the start flag to the final flag? Or 10 mins either side? Or the whole program? Or the start flag to the award of the trophy?

We coded the entire broadcast and have now clarified this in the methods section.

'These broadcasts were recorded in their entirety (start of the programme until the end of the credits), viewed and coded using the one-minute interval period method previously described (14-18)' – Page 5, line 10-12.

Might it be worth suggesting an alcohol equivalent of the FCTC?

This is a good suggestion and has now been added to the discussion section

'Whilst there is scope under Ofcom's powers to regulate content in programmes broadcast in the UK, Ofcom has no remit over sports sponsorship deals, and these are therefore a potentially unregulated source of exposure to alcohol content for young people. Furthermore, the F1 Championship races are broadcast internationally, which complicates regulating this content. The Framework Convention for Tobacco Control (42) is a treaty negotiated under the auspices of the World Health Organisation which provides a regulatory strategy to address addictive substances, has placed an international, comprehensive ban on the advertising, promotion and sponsorship of tobacco products (43). A similar comprehensive ban on the advertising, promotion and sponsorship of alcohol products would prevent young people being exposed to this currently unregulated alcohol promotion.' – Page 15, line 20 – Page 16, line 6.

Good Work- To be Commended

We thank you for this comment

VERSION 2 – REVIEW

REVIEWER	Pamela Trangenstein University of North Carolina at Chapel Hill
REVIEW RETURNED	05-May-2020
GENERAL COMMENTS	Nice job with the revisions!
REVIEWER	Frank Houghton Limerick Institute of Technology, Ireland.
REVIEW RETURNED	16-Apr-2020
GENERAL COMMENTS	Excellent piece of work.